# Meta-Analysis of the Impact of Far-Red Light on Vegetable Crop Growth and Quality

**DOI:** 10.3390/plants13172508

**Published:** 2024-09-06

**Authors:** Minggui Zhang, Jun Ju, Youzhi Hu, Rui He, Jiali Song, Houcheng Liu

**Affiliations:** College of Horticulture, South China Agricultural University, Guangzhou 510642, China; jetkiss@stu.scau.edu.cn (M.Z.); jujun-mail@stu.scau.edu.cn (J.J.); youzhihu@stu.scau.edu.cn (Y.H.); ruihe@stu.scau.edu.cn (R.H.); hxlsjl1046@scau.edu.cn (J.S.)

**Keywords:** far-red light, vegetable growth, morphological indicators, nutritional indicators, meta-analysis, statistical analysis

## Abstract

Far-red lights (FRs), with a wavelength range between 700 and 800 nm, have substantial impacts on plant growth, especially horticultural crops. Previous studies showed conflicting results on the effects of FRs on vegetable growth and quality. Therefore, we conducted a meta-analysis on the influence of FRs on vegetable growth, aiming to provide a comprehensive overview of their effects on the growth and nutritional indicators of vegetables. A total of 207 independent studies from 55 literature sources were analyzed. The results showed that FR treatment had significant effects on most growth indicators, including increasing the fresh weight (+25.27%), dry weight (+21.99%), plant height (+81.87%), stem diameter (+12.91%), leaf area (+18.57%), as well as reducing the content of chlorophyll (−11.88%) and soluble protein (−11.66%), while increasing soluble sugar content (+19.12%). Further subgroup analysis based on various factors revealed significant differences in the effects of FR on different physiological indicators, such as FR intensity, plant species, duration of FR exposure, and the ratio of red light to FR. In general, moderate FR treatment is beneficial for vegetable growth. This study provides important references and guidelines for optimizing the application of FR in the future.

## 1. Introduction

With population growth and climate change posing challenges to production, enhancing the efficiency, quality, and resilience of vegetable cultivation has become a pressing issue in agriculture. Artificial light sources can provide the necessary light energy for vegetables and influence their growth, yield, and quality through spectral and photoperiod adjustments. In this context, far-red light (FR), as a significant component of artificial lighting, presents broad prospects for application in plant growth regulation.

Far-red light (FR), situated at the boundary between the visible spectrum and infrared region of the electromagnetic spectrum, is an important component of the photosynthetic spectrum for plants. As a key element of the plant’s light environment, FR has a significant impact on the regulation of the growth cycle of vegetables [1,2]. FR is primarily involved in the photoperiodic effect, which is a physiological response of plants to changes in the length of daylight, affecting the flowering time, chlorophyll synthesis, and morphological parameters of plants [3,4,5], as well as influencing the biosynthesis of phytohormones and the regulation of plant growth [6]. The shade avoidance response is an adaptive growth reaction of plants to environments with a low red-to-far-red light ratio (R/FR), and this process is closely related to the photoreception of phytochromes [7]. Phytochrome exists in two forms within the plant: Pr (red-light-absorbing form) and Pfr (far-red-light-absorbing form). When plants are exposed to red light, Pr converts to Pfr, and FR promotes the Pfr reversion to Pr. This transformation between Pr and Pfr is the key mechanism by which plants sense the changes in photoperiods [8,9]. Under low R/FR conditions, the activation of phytochromes initiates the shade avoidance response, leading to a series of morphological changes in plants, such as internode elongation, leaf expansion, and petiole growth, to increase the capacity to capture light [10,11]. In protected cultivation, FR can be combined with other spectral components, such as red and blue light, to form the most favorable light environment for plant growth [12,13]; this comprehensive light quality management strategy not only improves crop yield and quality but also provides a new perspective for sustainable agricultural development.

Many studies have suggested that FR can significantly promote plant growth. The addition of FR to white light could increase the fresh and dry weight of lettuce but significantly reduce the content of pigments, total phenolic compounds, total flavonoid compounds, and vitamin C [14]. Exposure to FR could promote the height and stem diameter of tomato, cucumber, and pepper plants [15,16]. Conversely, some research suggested that FR might have negative effects on certain varieties of vegetables. The addition of far-red light to a white light background resulted in a decrease in both the fresh weight and dry weight of kohlrabi [17]. Furthermore, adding FR to the red and white composite light reduced the fresh weight and dry weight of edible ice plants, but it also promoted antioxidant activity and the biosynthesis of secondary metabolites [18]. Moreover, varying intensities and durations of FR treatments showed differences in vegetable growth. With the increasing intensity of FR, the fresh weight, dry weight, and leaf area of lettuce all exhibited a gradually increasing trend. Supplementing with an hour of far-red light at the end of the day significantly augmented the growth indices of lettuce [19]. As the ratio of red to far-red light (R/FR) decreased, the fresh weight and dry weight of the *Crepidiastrum denticulatum* plant significantly increased [20]. Low-intensity far-red light treatments effectively promoted the fresh weight, dry weight, leaf area, plant height, and internode length of Chinese kale and also increased the content of chlorophyll and carotenoids, although the leaf angle correspondingly became smaller [21]. Therefore, a comprehensive comparative analysis of the effects of various variables from numerous studies on FR will aid in understanding the impact of FR on vegetable growth.

Meta-analysis is a method that integrates results from multiple independent studies by systematically collecting, synthesizing, and analyzing existing research to derive more comprehensive and reliable conclusions [22]. This approach can bolster statistical power, unveil potential relationships, mitigate biases, and offer more reliable and persuasive evidence for scientific research [23]. Ma et al. conducted a meta-analysis that synthesized 695 independent studies from 56 published papers, delving into the effects of monochromatic and mixed-ratio LED lighting on various aspects of plant growth and development. They found that the family, growth cycle, and cultivation conditions of different plants significantly influenced the efficacy of LED lights [24]. Appolloni et al. conducted a meta-analysis that aggregated 100 independent studies from 31 published papers, thoroughly exploring the impact of supplementary LED lighting on greenhouse tomato production. It revealed the significant benefits of LED lighting in enhancing crop yield and improving fruit quality [25]. Badmus et al. conducted a meta-analysis that compiled 59 independent studies from 18 published papers, comprehensively assessing the influence of ultraviolet (UV) radiation on plant carotenoids, particularly its notable effect in increasing violaxanthin content. Additionally, it highlighted the potential multifaceted impacts of UV radiation on plant photosynthesis and photoprotection mechanisms [26]. By combining data from multiple studies, meta-analysis increases the sample size, improves statistical power, and enhances the reliability and persuasiveness of the results.

Therefore, we conducted a meta-analysis of papers published from 2014 to 2024 related to the regulation of vegetable growth and quality by FR to systematically summarize and evaluate the research findings reported in these publications.

## 2. Result

### 2.1. Overall Summary Effects

After conducting a comprehensive, integrated analysis of 207 independent studies, this research transformed the combined effect sizes of 27 physiological indicators into their corresponding natural logarithms and presented them visually in a forest plot (Figure 1). The magnitude of the effect size represents the degree of association between the study variables, and the dashed line at 0 is used to indicate whether the overall effect value is significantly different from zero. When the point of the combined effect size or the 95% confidence interval (CI) is on the dashed line at 0, it indicates that the FR treatment has no significant effect on this physiological indicator; it is statistically insignificant. When the point of the combined effect size is greater than 0, it indicates that the overall research results show a positive effect. Conversely, it indicates a negative effect.

From Figure 1, it can be seen that after assessing the 27 physiological indicators following FR treatment, 16 combined effect indicators had statistical significance. In terms of morphological indicators, such as fresh weight, dry weight, plant height, stem diameter, and leaf area, there was a positive impact, with increases of 25.17%, 21.99%, 81.87%, 12.91%, and 18.58%, respectively. Specific physiological and biochemical analysis indicators, such as SPAD and MDA, showed negative effects, with reductions of 14.79% and 14.21%, while the DPPH exhibited an increase of 4.19%, and the activities of superoxide dismutase (SOD) with reductions of 11.88%, while the activities of peroxidase (POD) and catalase (CAT) showed not significant effects.

Regarding nutritional indicators, primary metabolites like soluble protein showed a negative impact with a decrease of 17.68%. In contrast, the content of soluble sugar was positively affected, increasing by 19.12%. Nitrate content did not show significant changes. Chlorophyll content showed negative impacts, with reductions of 11.66%. Secondary metabolites, including carotenoids, Vitamin C, polyphenols, anthocyanins, and flavonoids, did not exhibit significant differences (*p* > 0.05).

### 2.2. Subgroup Analysis

To investigate the extent to which different factors affect the research outcomes, we conducted subgroup classifications to examine the impacts of different factors (FR intensity, crop family, photoperiod-EOD, and R/FR) on parameters such as fresh weight, dry weight, plant height, leaf area, chlorophyll content, soluble sugars content, and soluble protein content (Figure 2, Figure 3, Figure 4, Figure 5 and Figure 6, Table 1 and Table 2).

#### 2.2.1. The Influence of FR Intensity

The study was divided into three subgroups based on the intensity of FR: low intensity (0–50 μmol·m^−2^·s^−1^), medium intensity (50–100 μmol·m^−2^·s^−1^), and high intensity (>100 μmol·m^−2^·s^−1^). In terms of fresh weight, the effects of promoting plant growth gradually increased with the increasing FR intensity. Specifically, in the low-intensity group, the overall was 0.20, while in the high-intensity group, the overall significantly increased to 0.37, indicating a 44.63% increase in fresh weight. In addition, dry weight, plant height, and leaf area also showed similar trends, with the overall effect sizes reaching 0.31, 0.34, and 0.81 in the high-intensity group, corresponding to growth rates of 35.70%, 125.08%, and 41.06%, respectively. However, there was a negative inhibitory effect on chlorophyll content as light intensity increased, especially in the high-intensity group, where the chlorophyll content significantly decreased by 32.64%. The soluble sugar content in the medium-intensity group increased by 22.74%, while the soluble protein content in the low-intensity group decreased by 26.64%.

#### 2.2.2. The Influence of Vegetable Family

Based on the family of vegetables, they were categorized into Asteraceae, Brassicaceae, Lamiaceae, Solanaceae, Cucurbitaceae, and others. The results indicated that there were positive promoting effects of FR on plant height of the Cucurbitaceae and Solanaceae, with an increase of 278.22% and 85.10%, respectively. In contrast, soluble protein content was reduced by 50.66% and 36.06%, respectively. Additionally, there were small but significant promoting effects of FR on plant height in other families. In Brassicaceae and Asteraceae, fresh weight, dry weight, leaf area, plant height, and soluble sugar content responded positively to FR, and the fresh weight, dry weight, and soluble sugar content positively increased by 17.72%, 38.76%, and 23%, respectively, while the chlorophyll content and soluble protein content decreased by 11.21% in the Brassicaceae. In the Asteraceae, the dry weight and soluble sugar content positively increased by 19.87% and 17.99%, and chlorophyll content decreased by 12.52%. There was no significant effect on the leaf area, while there was a 20.48% increase in dry weight in the Lamiaceae by FR.

#### 2.2.3. The Influence of the Photoperiod with FR Supplementation

The photoperiod with FR supplementation was divided into 12 h, 16 h, >16 h, and EOD (end of day) subgroups. The results showed that all subgroups generally exhibited a positive promoting effect on fresh weight and dry weight, with the 12 h subgroup having the highest overall effect on fresh weight, dry weight, and plant height, which were 0.3034, 0.3053, and 0.76, respectively, corresponding to an increase of 35.45%, 35.70%, and 114.88%. For the leaf area, the EOD subgroup had the highest overall effect of 0.21, with an increase of 23.96%. However, chlorophyll content showed inhibitory effects across all subgroups. Particularly, the 16 h subgroup had an overall effect size of −0.3514, with a negative decrease of −29.63%. The soluble sugar content in the 12 h group and the EOD group increased by 22.79% and 13.10%, respectively, while the soluble protein content decreased by 24.63% and 10.21%, respectively.

#### 2.2.4. The Influence of the Red to FR Ratio (R/FR)

Based on the ratio of red to FR (R/FR), the groups were divided into a low R/FR ratio range (0–1), medium R/FR ratio range (1–2), high R/FR ratio range (2–4), very high R/FR ratio range (4–6), and extremely high R/FR ratio range (6, ∞). The results showed that when the R/FR ratio decreased, the overall effects sizes for fresh weight, dry weight, and leaf area were enhanced, with more significant positive promoting effects. For fresh weight, the overall effect size in the (0–1) range reached 0.3583, with a positive increase of up to 43.09%. When the R/FR ratio increased, the overall effect size declined, but the effect size and change rate remained significantly positive. However, in all R/FR ratio ranges, the chlorophyll content showed a negative overall effect, with the lower R/FR ratio range (0–1) having a chlorophyll overall effect size of −0.2047, with a negative decrease of 18.51%. The soluble sugar content in the (2–4) range increased by 22.34%, while the soluble protein content decreased by 23.70%.

### 2.3. Heterogeneity Analysis

To gain a deeper understanding of the differences between various study results and the potential impact of these differences on the interpretation of the overall effect, this study employed the P_-hetero_ value of the Q statistic to assess whether there was significant heterogeneity in the effects of FR treatment on plant physiological indicators. The I^2^ statistic was used to describe the degree of heterogeneity, as shown in Table 3. The results indicated that out of the 27 physiological indicators studied, 8 showed heterogeneity (P_-hetero_ < 0.1). Specifically, fresh weight and leaf area exhibited low heterogeneity (I^2^ ≤ 25%), while SPAD, the content of chlorophyll, soluble protein, and polyphenols showed moderate heterogeneity (25% < I^2^ ≤ 50%), and carotenoid content and DPPH showed high heterogeneity (I^2^ > 50%).

In this study, considering that the sample size directly affects the reliability of the statistical analysis, we chose to conduct a detailed subgroup analysis for the indicators with larger sample sizes, namely the fresh weight and leaf area (Figure 2 and Figure 4). The results showed that the heterogeneity in fresh weight was primarily distributed in subgroups such as the Brassicaceae family, a 12 h photoperiod, and the R/FR ratio interval (1,2). For the leaf area, despite subgroup analysis, significant heterogeneity remained among different subgroups, with the I^2^ values for the Brassicaceae and Asteraceae families being 41.27% and 25.46%, respectively, indicating moderate heterogeneity.

### 2.4. Publication Bias

In the execution of a meta-analysis, the evaluation of publication bias is instrumental in safeguarding the accuracy, credibility, and objectivity of the research findings. Upon conducting an analysis of the indicators with significant effects using funnel plots and linear regression tests (Appendix A: ‘Publication bias’ sheet), it was observed that there may have been a potential for publication bias in the indicators of stem diameter and soluble sugar content (as indicated by a *p*-value from the Egger’s test that is less than 0.05). The remaining indicators, however, showed a better performance with respect to publication bias, with no significant signs of such biases detected.

## 3. Discussion

### 3.1. Research Statement and Summary Analysis

This study aimed to investigate the impacts of FR on the growth and quality of vegetables. Consequently, a meta-analysis of a substantial body of research was conducted, focusing on the effects of FR on the morphological indicators and nutritional quality indicators of vegetables. A total of 55 papers were ultimately included in the meta-analysis. Within the entire dataset, research from 11 countries contributed to this topic, with 87% of the studies originating from five countries, indicating a significant concentration of research in the field of FR regulation of vegetable growth in these nations, such as China, the United States, Republic of Korea, the Netherlands, and Japan (Figure 7). Among the 206 independent studies identified, Asteraceae vegetables accounted for the largest proportion at 38.3%, while Lamiaceae vegetables constituted only 6.8% (Figure 8).

FR had a significant promoting effect on the morphological indicators of vegetables [27,28]. The application of FR increased the yield and plant height of vegetables and also enhanced the photosynthetic efficiency [29]. Under R/FR = 3.81, the dry weight and leaf area of lettuce increased by 46–77% and 58–75%, respectively, with an increase in the incident light utilization efficiency by 17–42% [27]. Additionally, the supplementation of FR on the white light increased the fresh weight of ‘Red butter’ and ‘Green butter’ lettuce by 35.68% and 37.09%, respectively [30]. The supplementation of 30 μmol·m^−2^·s^−1^ of FR significantly enhanced the fresh weight of cucumber and tomato seedlings (by 102.78% and 43.88%, respectively) as well as the plant height (by 194.47% and 124.79%, respectively) [15]. The meta-analysis results showed that the supplementation of FR promoted the morphological growth of vegetables (Figure 1), particularly in aspects such as plant height, stem diameter, and hypocotyl length. However, the impact of FR on the number of leaves and specific leaf area of vegetables was not significant, possibly because the simultaneous increase in leaf area and dry weight may have offset each other to some extent. The increase in leaf biomass might not be accompanied by corresponding morphological adjustments, or the rate of leaf expansion might be comparable to the rate of biomass accumulation, resulting in no significant difference in the specific leaf area index [31]. Further subgroup analysis (see Figure 5, Figure 6, Figure 7, Figure 8 and Figure 9 for details) showed that with the increase in FR intensity or the decrease in the R/FR ratio, the effects of increasing fresh weight, dry weight, and leaf area of vegetables became more pronounced. Different taxonomic families of vegetables showed a certain degree of increased fresh and dry weight after FR treatment. Compared to the vegetables of Asteraceae, Brassicaceae, Solanaceae, and Cucurbitaceae families, the leaf area growth effect of Lamiaceae vegetables was not significant (*p* > 0.05), which might be due to insufficient research data on leaf areas in this family, leading to an increased 95% confidence interval of the effect value [32,33]. The effects of FR supplementation before the end of the day on the increased fresh weight, dry weight, plant height, and leaf area of vegetables were also very significant [30,34,35], with the increase in fresh weight of 25.71%, dry weight of 25.71%, and leaf area of 23.96%, indicating that FR supplementation before the end of the day also had significant impacts on vegetable growth. However, only five papers were included in the study regarding FR supplementation before the end of the day, and further exploration is needed in the future.

FR also had significant impacts on the nutrition of vegetables [36,37]. On the basis of R:B = 7:1, the supplementation of FR before the end of the day increased the content of soluble sugars and nitrates in lettuce but reduced the soluble protein content [19]. The supplementation of 10 μmol·m^−2^·s^−1^ FR increased the content of soluble sugars and soluble protein in lettuce [14], whereas the supplementation of 50 μmol·m^−2^·s^−1^ FR reduced the content of soluble sugars and soluble protein in lettuce [38]. Consistently, the content of chlorophyll and carotenoids decreased. In this dataset, the indicators that have a significant impact on the content of soluble sugars and soluble protein were mainly focused on Asteraceae and Brassicaceae vegetables. Further subgroup analysis showed (see Table 1 and Table 2) that the supplementation of FR to vegetables could increase soluble sugar content and reduce soluble protein content. For other nutritional indicators such as vitamin C, polyphenols, and flavonoids, due to the limited number of related studies, it is difficult to perform a subgroup analysis, which, in turn, limits the accurate assessment of these effects. In studies with small samples, the results of individual studies are more susceptible to random factors, and statistical analysis might lack sufficient power to detect true effects [22,39]. For instance, there were nine independent studies analyzing the vitamin C content, but the subjects of these nine independent studies were all lettuce, which did not effectively evaluate the impact of FR on the vitamin C content of other varieties, such as Brassicaceae and Cucurbitaceae vegetables. Therefore, the current research does not have enough data to analyze the impact of FR on these nutritional indicators, and fewer studies have measured the antioxidant capacity indicators. Thus, future research could expand on the range of study subjects to include more types of vegetables, especially those that are commonly consumed in daily diets. By studying multiple vegetable varieties, a more comprehensive understanding of the impact of FR on the nutritional indicators of different types of vegetables could be obtained.

### 3.2. The Accuracy of Data

The presence of heterogeneity might compromise the reliability and stability of the meta-analysis results. In Section 2 of this paper, indicators such as SPAD, chlorophyll, soluble protein, polyphenols, carotenoids, and DPPH all exhibited moderate to high heterogeneity, with significant differences observed in chlorophyll, soluble protein, and DPPH. Subgroup analysis of the chlorophyll content (Figure 6) revealed that subgroups with an extremely high R/FR ratio (6, ∞), a 12 h FR photoperiod, Cruciferous vegetables, and low-intensity FR exhibited significant heterogeneity (I^2^ values are 88.21%, 51.28%, 64.65%, and 55.25%, respectively). The subgroup analysis of soluble proteins also showed a similar trend (Table 3). These findings suggested that when planning future experiments, it was essential to exert more precise control over experimental conditions. For instance, the accurate regulation of the photoperiod and the ratio of light quality might significantly affect the chlorophyll content. Moreover, the growth characteristics of different vegetable families and their sensitivity to environmental changes are also factors that should not be overlooked in experimental design. Therefore, future research should adopt more rigorous experimental design methods aimed at reducing potential confounding factors, thereby enhancing the comparability and reliability of the research results. Through this strategy, we can more accurately assess the impact of FR on plant physiological traits and provide a solid foundation for research in related fields. The small sample size involved in the analysis of the DPPH indicator might lead to an unstable estimation of heterogeneity for this indicator, increasing the uncertainty of the results. A small sample size might amplify the impacts of chance factors, thereby masking or exaggerating the actual heterogeneity present [40].

In the publication bias analysis, there was evidence of publication bias in the indicators of stem diameter and soluble sugar (Appendix A ‘Publication bias’ sheet). This implies a certain degree of selective reporting of results. However, the majority of the independent studies (46 studies on stem diameter accounting for 76.08%, and 25 studies on leaf area accounting for 88%) supported the conclusion that the addition of FR promoted the stem diameter and soluble sugar content in vegetables. This consistency significantly enhanced the reliability of the conclusion. Even if there may have been some publication bias, there was strong evidence supporting the promoting effect of added FR on the stem diameter and soluble sugar content [41].

### 3.3. Limitations and Recommendations

This study lacked an in-depth exploration of the biological mechanisms behind vegetable growth, which might limit a comprehensive understanding of the impacts of FR and thus affect the interpretation and inference of research results. Additionally, due to the limited studies on nutritional indicators, it was difficult to fully evaluate the effects of FR on vegetable quality, which might weaken the reliability and practicality of the research results. Furthermore, there might be potential issues with sub-group classification. Due to the limited research on the Solanaceae and Cucurbitaceae families, merging them for subgroup analysis could obscure the growth differences between them, leading to bias in the study conclusions.

In future research, the following areas could be expanded. Firstly, there should be in-depth research on the biological mechanisms of FR on vegetable growth, including research on photosynthesis, phytohormone, and gene expression to reveal the pathways through which FR affects vegetable growth. Secondly, the scope of research on nutritional indicators should be broadened to include diverse indicators such as the photosynthetic rate, enzyme activity, and root structure, among others, to comprehensively evaluate the effects of FR on vegetable growth. Simultaneously, exploring the effects of FR on vegetable growth under different planting conditions, including factors such as light intensity, photoperiod, temperature, and humidity, is necessary to determine the optimal planting conditions. Additionally, studying the interactions between FR and other environmental factors (such as CO_2_ concentration and water supply) to find the best application strategies of FR under optimal planting conditions is crucial. Furthermore, investigating the impact of FR on plant stress resistance, including droughts, high temperatures, and salinity, as well as the potential mechanisms by which FR improves vegetable resilience, is needed. Further exploration of the recovery effects of FR pre-treatment on plant stress after adversity is essential to enhance the stability of vegetable yields.

## 4. Materials and Methods

### 4.1. Data Collection

In this study, we conducted searches through PubMed and the Web of Science databases, obtaining a total of 583 relevant papers (the search keywords are listed in Appendix A: ‘Search query’ sheet). We only collected accessible English-language materials, including scientific articles, conference papers, book chapters, and dissertations. We then filtered the search results to reduce heterogeneity in the study. After a series of screenings and checks, we excluded 552 papers that did not meet the content requirements of the study (Figure 9). Ultimately, 55 papers on the impact of FR on vegetable growth and quality were included in the Meta-analysis (detailed information is provided in Appendix A ‘Included literature’ sheet), with the publication years of relevant academic papers from around the world mainly concentrated between 2016 and 2024 (Figure 7 and Figure 10). This study primarily explores the impact of FR on the growth and quality of vegetables, and other light quality ratio factors are not included in the scope of influence. Therefore, the control group in the study consists of all light treatments mentioned in the articles, including white light (W), red-blue light (RB), red-white light (RW), and red-blue-white light (RBW), etc. The treatment group, on the other hand, is based on the addition of FR to the corresponding control group. If a paper contains multiple treatments, each treatment is considered an independent study, and data from the control group will be used multiple times to calculate effect values [42,43].

Extracted numerical data were directly taken from the original literature or tables, while graphical data were extracted using GetData 0.11.0 (https://getdata.sourceforge.net/). The extracted data were used to establish a database in Microsoft Excel 2021, which includes the following categories: authors, publication years, titles, vegetable varieties, light treatments for the treatment and control groups, the intensity of added FR, the intensity of red light in the treatment group and the ratio of red light to far-red light, photoperiod, and the mean, standard deviation (SD) and sample size (n) for 27 physiological indicators for both the control and treatment groups. If the study only provides the standard error (SE), it is converted to standard deviation (SD) using the formula:(1)$$\mathrm{SD}=SE\times \sqrt{n}$$

### 4.2. Categorical Variable

In meta-analysis, subgroup analysis is a method that classifies research subjects according to a specific variable and then analyzes and compares different categories to explore the extent of the impact of various factors on the research results [44,45]. By analyzing each subgroup, it is possible to understand whether there are differences in the growth effects of FR on different types of vegetables and whether the effects of FR differ under different treatment conditions.

We conducted subgroup analyses on four variables that may affect the growth of vegetables by FR from each study, which include (I) the intensity of FR, (II) the botanical family of the plants, (III) the irradiation cycle of FR, and (IV) the ratio of red to FR, denoted as R/FR. The effect value data for each variable must come from at least two independent studies and cover two different publications [46]. All these variables will be used to analyze and explain the differences in the overall effect values that have significant effects in the meta-analysis [47].

### 4.3. Meta Analysis

From the 55 papers included, we extracted a total of 207 independent studies and included 27 physiological indicator effect parameters, which are commonly used to evaluate the degree of response of vegetables to FR irradiation. The data in this study belong to continuous variables, and a random-effects model was chosen for analysis. Data processing was performed using the MetaWin 3 software for analysis [48], and the forest plot was drawn using the software Origin 2021. We adopted the natural logarithm of the mean ratio of physiological indicators (lnR), that is, the natural logarithm of the ratio of the average values of the treatment group and the control group, as the effect size and calculated its variance.
(2)$$\ln R=\ln(\frac{Xt}{Xc})$$
(3)$$Var=\frac{SD_{t}^{2}}{n_{t}X_{t}^{2}}+\frac{SD_{c}^{2}}{n_{c}X_{c}^{2}}$$

In this context, R represents the response rate, a unitless parameter that indicates the relative change in the treatment group compared to the control group. This indicator can be used to measure the overall effect size and is commonly used to assess the overall effect of all included studies [24]. X_t_ is the mean of the FR treatment group, and X_c_ is the mean of the control group. The natural logarithm of the effect size reflects the relative magnitude of the treatment effect [49]. n_t_ and n_c_ are the number of replicates in the treatment and control groups, respectively. SD_t_ and SD_c_ are the standard deviations of the treatment and control groups, respectively.

Each physiological indicator must be supported by at least two pieces of literature before it can be included in the calculation of the overall effect size. The use of the natural logarithm transformation helps to normalize the data distribution, thereby simplifying subsequent statistical analysis. It also reduces the disparity between data points, making the data from different groups more comparable, and can mitigate the impact of extreme values on the results, enhancing the stability and reliability of the findings [46,50]. An lnR value greater than 0 indicates that the measured physiological indicator has increased after FR treatment, an lnR value less than 0 indicates a decrease in the measured physiological indicator, and an lnR value equal to 0 indicates no change in the measured physiological indicator. To better understand, the effect size can be converted into a percentage change through the following formula:(4)$$\mathrm{Change}=(e^{\ln R}-1)\times 100\%$$

We used sampling methods to assess whether the study results were significant. If the two-tailed *p*-value of the z-test was less than the level of significance (such as 0.05), we could reject the null hypothesis and consider the overall effect to be significant. Otherwise, there was no significance [51]. Heterogeneity indicated that there were significant differences between different studies, which meant that the variability in study results exceeded the range of random error [52]. In meta-analysis, the Q statistic is commonly used to measure the heterogeneity between different study results. The P_-hetero_ value represents the probability of observing the sample data or more extreme cases under the null hypothesis. I^2^ is a statistical measure that describes the degree of heterogeneity between studies, indicating the percentage of total heterogeneity caused by true differences [53,54]. If the *p*-value is small, usually less than 0.10, or I^2^ is a positive value (I^2^ < 25% is low heterogeneity, 25% ≤ I^2^ < 50% is moderate heterogeneity, I^2^ ≥ 50% is high heterogeneity), we reject the null hypothesis, indicating that heterogeneity exists [55,56], and this situation requires careful interpretation of the meta-analysis results.

Publication bias refers to the phenomenon where significant or positive results are more likely to be published, while non-significant or negative results are often ignored, thereby affecting the conclusions of the meta-analysis [57]. Common methods to detect and correct publication bias include the funnel plot, linear regression test (Egger’s regression test), Begg’s test, and the Trim and Fill method [58]. The funnel plot is one of the most commonly used tools, which can intuitively display the symmetry of the study results by plotting the relationship between the standard error of the study results and the effect size. If publication bias exists, the funnel plot will show an asymmetrical shape [59]. The linear regression test is also a common method, as is Egger’s regression test. By performing a linear regression analysis on the points in the funnel plot, this method can quantify the degree of bias [60]. Begg’s test can statistically assess the presence of publication bias. The Trim and Fill method is a way to estimate the true effect size more accurately by trimming and filling operations on the data [61]. These methods are usually used in combination to comprehensively assess the impact of publication bias on the study results, thereby ensuring the accuracy and reliability of the meta-analysis.

## 5. Conclusions

This study systematically explored the effects of FR on the growth and quality of vegetables through a meta-analysis. The results demonstrated that FR treatment significantly influenced the morphological and nutritional indices of vegetables, including increased fresh weight, dry weight, plant height, and soluble sugar content, while the treatment decreased the contents of chlorophyll and soluble protein. Further subgroup classification analysis revealed significant differences in the impact of factors, such as FR intensity, vegetable family, photoperiod, and the R/FR ratio, on various physiological indices. The stability and reliability of the research findings were verified through a publication bias analysis. In summary, this study provides important insights into the role of FR in vegetable growth and offers guidance for future related research.

## Figures and Tables

**Figure 1 plants-13-02508-f001:**
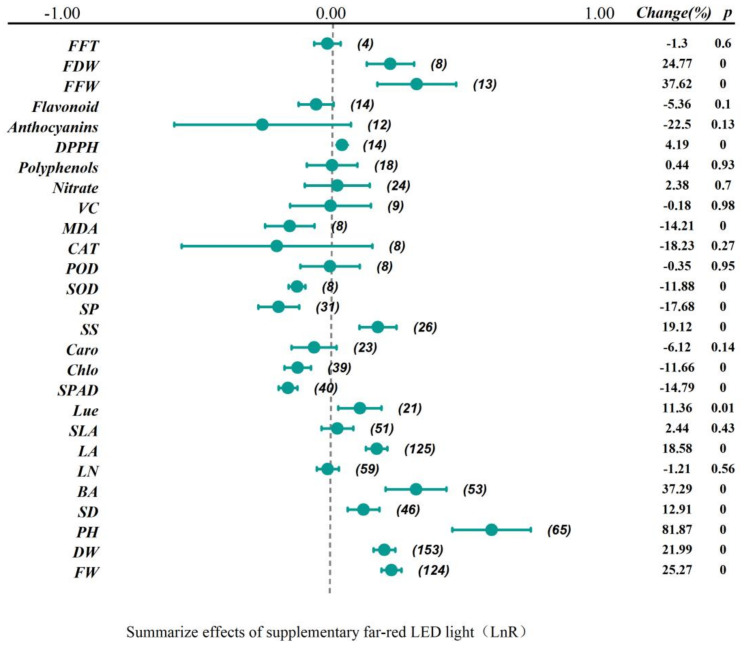
The forest plot displays the response ratios of various physiological indicators of vegetables to FR (FFT: First Flowering Time. FDW: Fruit Dry Weight. FFW: Fruit Fresh Weight. DPPH: DPPH Radical Scavenging Rate. VC: Vitamin C. MDA: Malondialdehyde. CAT: Catalase. POD: Peroxidase. SOD: Superoxide Dismutase. SP: Soluble Protein. SS: Soluble Sugar. Caro: Carotenoids. Chlo: Chlorophyll. SPAD: Relative Chlorophyll Content. Lue: Light Energy Utilization Efficiency. SLA: Specific Leaf Area. LA: Leaf Area. LN: Leaves Number. HL: Hypocotyl Length. SD: Stem Diameter. PH: Plant Height. DW: Dry Weight. FW: Fresh Weight), along with the main analysis parameters. The numbers in parentheses refer to the sample sizes of independent studies. The parameters of the meta-analysis include the size of the effect size, 95% CI, the percentage change rate, and the zero-hypothesis test (two−tailed *p*-value).

**Figure 2 plants-13-02508-f002:**
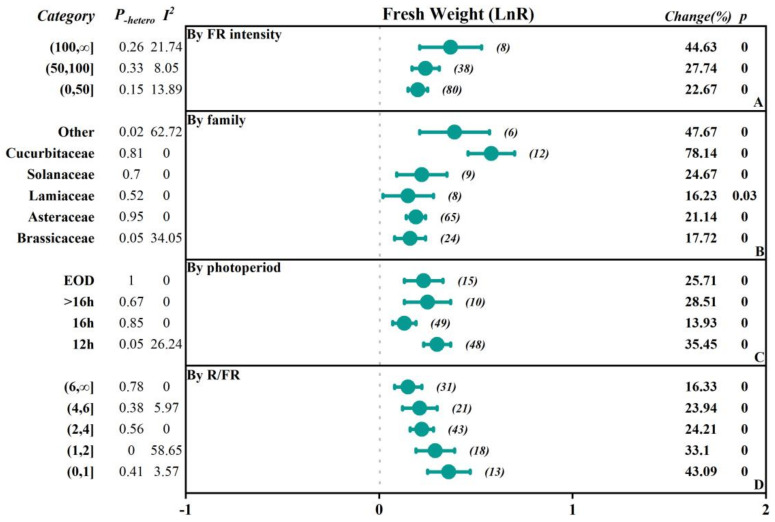
Subgroup Analysis Forest Plot illustrates the overall effect size of the response ratio of FR on the fresh weight of vegetables and the main analysis parameters. The numbers in parentheses refer to the sample size of independent studies. Parameters for meta-analysis include the effect size, 95% confidence interval (CI), percentage change rate, Q-test *p*-value for heterogeneity (P_-hetero_), I^2^ statistic, and the null-hypothesis test (two-sided *p*-value).

**Figure 3 plants-13-02508-f003:**
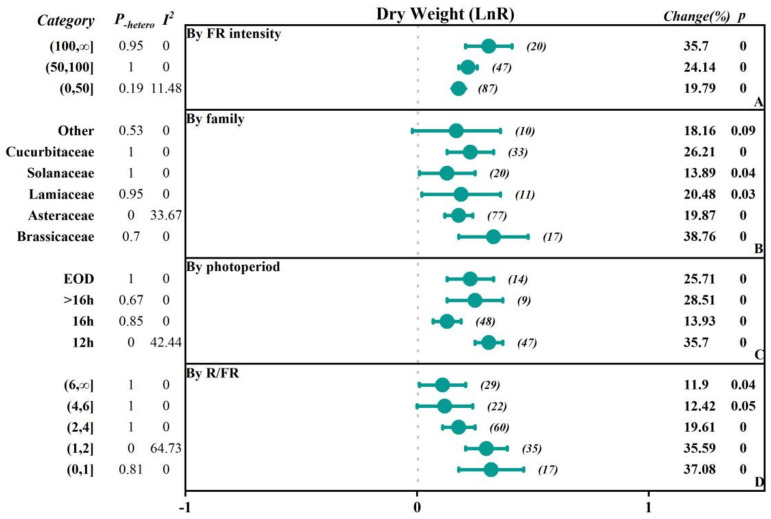
Subgroup Analysis Forest Plot illustrates the overall effect size of the response ratio of FR on the dry weight of vegetables and the main analysis parameters. The numbers in parentheses refer to the sample size of independent studies. Parameters for meta-analysis include the effect size, 95% confidence interval (CI), percentage change rate, Q-test *p*-value for heterogeneity (P_-hetero_), I^2^ statistic, and the null-hypothesis test (two-sided *p*-value).

**Figure 4 plants-13-02508-f004:**
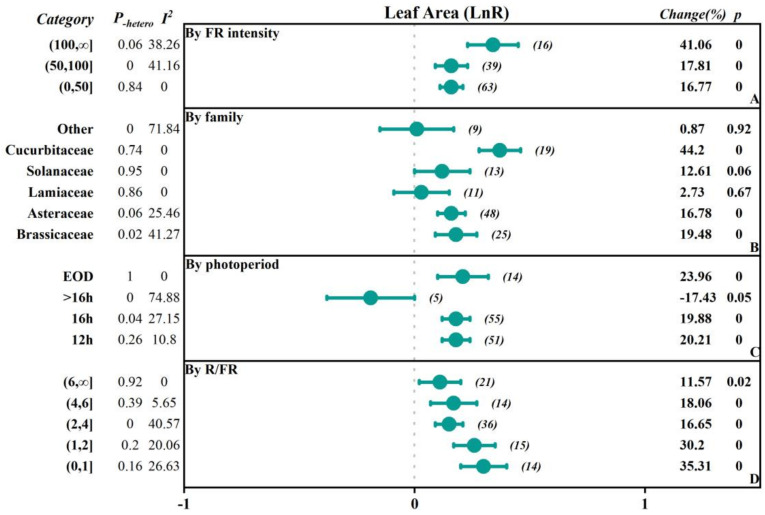
Subgroup Analysis Forest Plot illustrates the overall effect size of the response ratio of FR on the leaf area of vegetables and the main analysis parameters. The numbers in parentheses refer to the sample size of independent studies. Parameters for meta-analysis include the effect size, 95% confidence interval (CI), percentage change rate, Q-test *p*-value for heterogeneity (P_-hetero_), I^2^ statistic, and the null-hypothesis test (two-sided *p*-value).

**Figure 5 plants-13-02508-f005:**
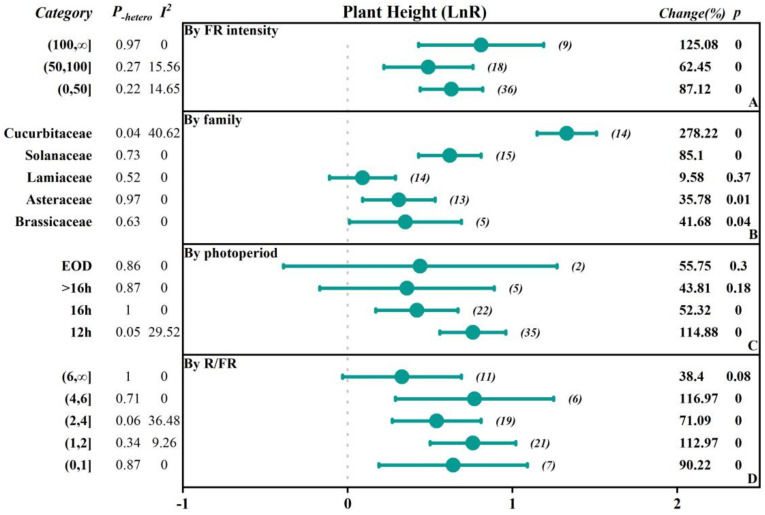
Subgroup Analysis Forest Plot illustrates the overall effect size of the response ratio of FR on the plant height of vegetables and the main analysis parameters. The numbers in parentheses refer to the sample size of independent studies. Parameters for meta-analysis include the effect size, 95% confidence interval (CI), percentage change rate, Q-test *p*-value for heterogeneity (P_-hetero_), I^2^ statistic, and the null-hypothesis test (two-sided *p*-value).

**Figure 6 plants-13-02508-f006:**
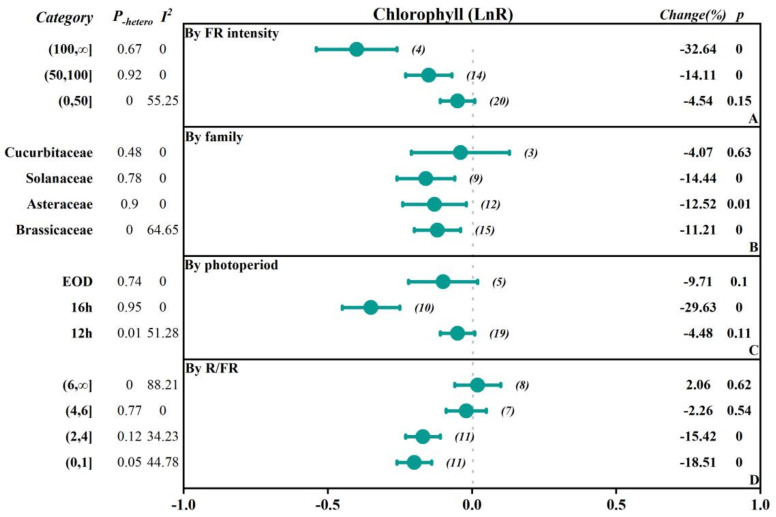
Subgroup Analysis Forest Plot illustrates the overall effect size of the response ratio of FR on the chlorophyll content of vegetables and the main analysis parameters. The numbers in parentheses refer to the sample size of independent studies. Parameters for meta-analysis include the effect size, 95% confidence interval (CI), percentage change rate, Q-test *p*-value for heterogeneity (P_-hetero_), I^2^ statistic, and the null-hypothesis test (two-sided *p*-value).

**Figure 7 plants-13-02508-f007:**
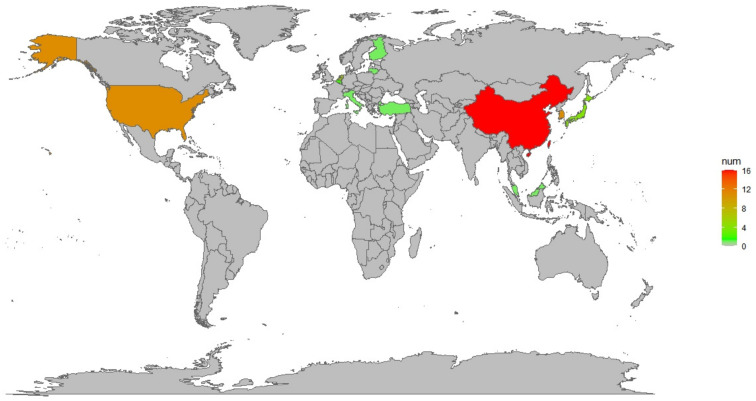
The global geographical distribution of the 55 research articles included in the analysis.

**Figure 8 plants-13-02508-f008:**
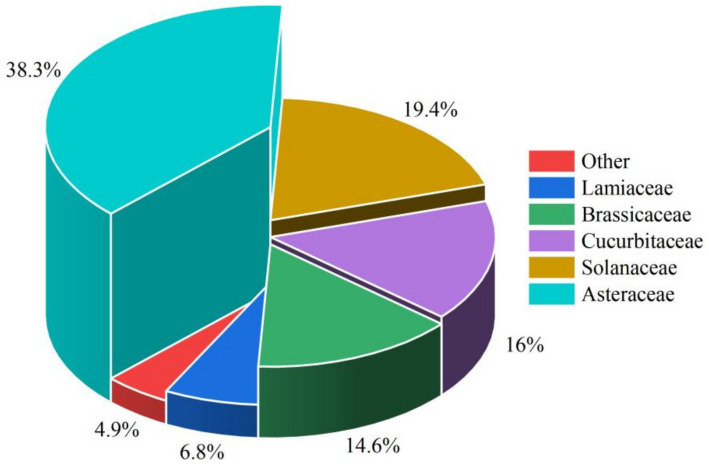
Different proportions of research on vegetables from various families.

**Figure 9 plants-13-02508-f009:**
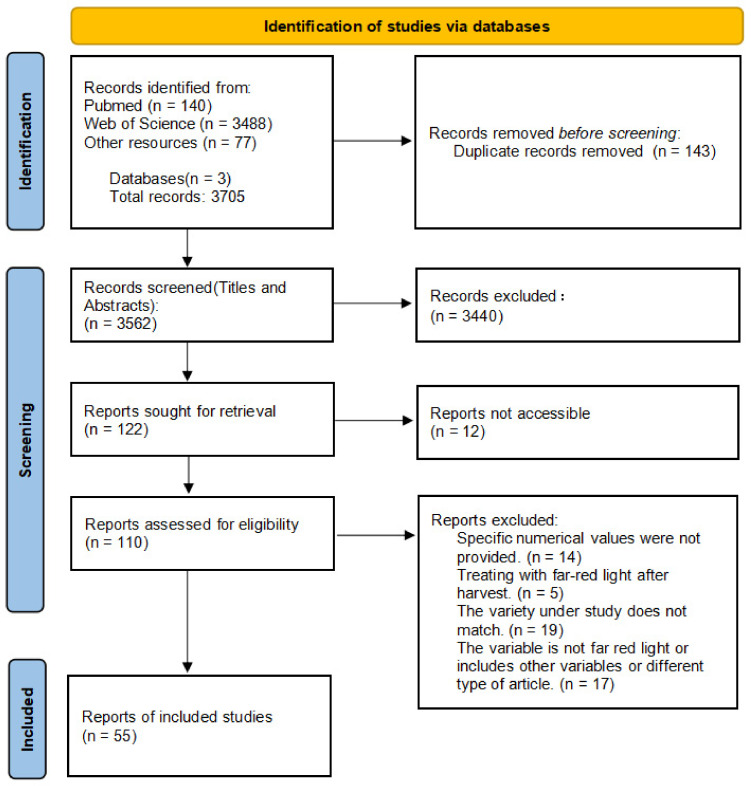
The PRISMA flow diagram systematically reviews the process of literature screening for meta-analysis. It illustrates the number of identified records, the number of included and excluded records, as well as the reasons for exclusion.

**Figure 10 plants-13-02508-f010:**
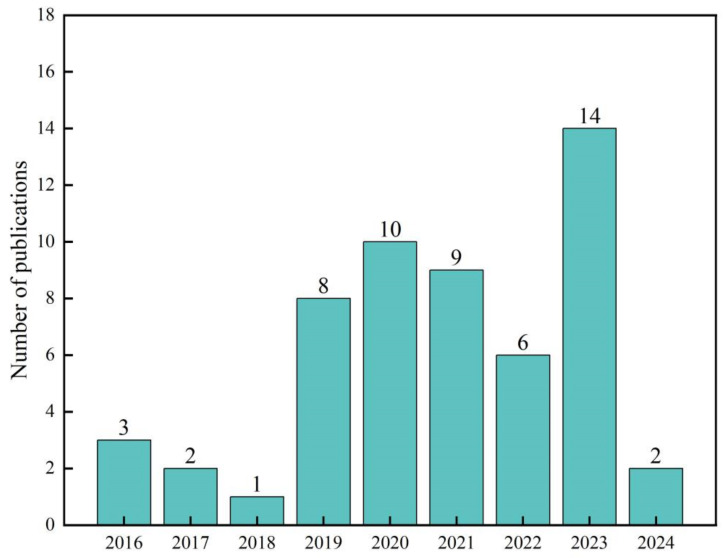
The cumulative number of articles published annually from January 2014 to April 2024.

**Table 1 plants-13-02508-t001:** Subgroup analysis tables demonstrate the impact of FR treatment on the main analytical parameters of soluble sugar content in vegetables.

Subgroup Factors	No. of Studies	lnR (95%CI)	*p*	Heterogeneity		Change (%)
				P_-hetero_	I^2^	
Total	26	0.18 (0.11, 0.24)	0.00	0.24	15.64	19.12
R/FR						
(0,1]	8	0.21 (0.08, 0.34)	0.00	0.98	0.00	23.07
(1,2]	2	0.13 (−0.13, 0.38)	0.33	0.76	0.00	13.53
(2,4]	9	0.2 (0.07, 0.33)	0.00	0.28	17.75	22.34
(4,6]	3	−0.03 (−0.25, 0.19)	0.78	0.14	48.74	−3.11
(6,∞]	3	0.2 (−0.07, 0.46)	0.14	0.15	47.56	21.59
Photoperiod						
12 h	10	0.21 (0.12, 0.29)	0.00	0.69	0.00	22.79
EOD	10	0.12 (0.04, 0.21)	0.00	0.06	44.99	13.10
Family						
Brassicaceae	8	0.21 (0.08, 0.26)	0.00	0.97	0.00	23.00
Asteraceae	17	0.17 (0.08, 0.33)	0.01	0.05	38.85	17.99
FR intensity						
(0,50]	6	0.06 (−0.1, 0.22)	0.47	0.07	51.00	6.08
(50,100]	20	0.2 (0.13, 0.28)	0.00	0.68	0.00	22.74

**Table 2 plants-13-02508-t002:** Subgroup analysis tables demonstrate the impact of FR treatment on the main analytical parameters of soluble protein content in vegetables.

Subgroup Factors	No. of Studies	lnR (95%CI)	*p*	Heterogeneity		Change (%)
				P_-hetero_	I^2^	
Total	124	−0.19 (−0.27, −0.12)	0.00	0.09	15.09	−17.68
R/FR						
(0,1]	8	−0.05 (−0.19, 0.09)	0.48	0.99	0.00	−4.92
(2,4]	10	−0.27 (−0.41, −0.14)	0.00	0.03	52.65	−23.70
(4,6]	8	−0.36 (−0.51, −0.21)	0.00	0.01	63.14	−30.36
(6,∞]	4	−0.04 (−0.24, 0.17)	0.73	0.37	4.70	−3.49
Photoperiod						
12 h	16	−0.28 (−0.39, −0.17)	0.00	0.00	60.20	−24.63
EOD	9	−0.11 (−0.25, 0.03)	0.14	1.00	0.00	−10.21
Family						
Brassicaceae	9	−0.08 (−0.17, 0.02)	0.12	0.85	0.00	−7.27
Asteraceae	14	−0.09 (−0.17, 0)	0.04	0.78	0.00	−8.27
Solanaceae	5	−0.45 (−0.59, −0.31)	0.00	0.00	84.46	−36.06
Cucurbitaceae	3	−0.71 (−0.89, −0.52)	0.00	0.06	65.19	−50.66
FR intensity						
(0,50]	15	−0.31 (−0.41, −0.21)	0.00	0.00	72.42	−26.64
(50,100]	16	−0.09 (−0.18, 0.01)	0.07	1.00	0.00	−8.21

**Table 3 plants-13-02508-t003:** Heterogeneity analysis of the overall effect sizes following FR treatment for 20 indicators. The *p*-value indicates the probability of observing the data under the null hypothesis (two-tailed *p*-value). The P_-hetero_ value represents the probability of observing the sample data or a more extreme scenario if the null hypothesis is true. It is used to assess the heterogeneity among studies. I^2^ is a statistical measure that quantifies the degree of heterogeneity between studies, expressing the percentage of total variability that is attributable to true differences rather than random error.

Summarize Effects Size	No. of Studies	ln R (95%CI)	*p*	Heterogeneity	
				P_-hetero_	I^2^
Fresh Weight	124	0.23 (0.19, 0.26)	0.00	0.09	15.09
Dry Weight	168	0.2 (0.16, 0.24)	0.00	0.20	8.28
Plant Height	65	0.6 (0.45, 0.74)	0.00	0.47	0.09
Stem Diameter	46	0.12 (0.06, 0.18)	0.00	0.34	6.81
Hypocotyl Length	53	0.32 (0.2, 0.43)	0.00	0.24	11.58
Leaf Number	59	−0.01 (−0.05, 0.03)	0.56	0.86	0.00
Leaf Area	125	0.17 (0.13, 0.21)	0.00	0.01	24.00
Specific Leaf Area	51	0.02 (−0.04, 0.08)	0.43	0.70	0.00
Light Use Efficiency	21	0.11 (0.03, 0.19)	0.01	0.85	0.00
SPAD	40	−0.16 (−0.2, −0.12)	0.00	0.04	30.75
Chlorophyll	39	−0.12 (−0.17, −0.08)	0.00	0.02	34.62
Carotenoids	23	−0.06 (−0.15, 0.02)	0.14	0.00	66.90
Soluble Sugars	26	0.18 (0.11, 0.24)	0.00	0.24	15.64
Soluble Protein	31	−0.19 (−0.27, −0.12)	0.00	0.01	43.23
SOD	8	−0.13 (−0.16, −0.1)	0.00	0.15	34.31
POD	8	0 (−0.11, 0.11)	0.95	0.21	27.96
CAT	8	−0.2 (−0.56, 0.15)	0.27	0.31	14.94
MDA	8	−0.15 (−0.25, −0.06)	0.00	0.49	0.00
Vitamin C	9	0 (−0.15, 0.15)	0.98	0.49	0.00
Nitrate	24	0.02 (−0.1, 0.15)	0.70	0.95	0.00
Polyphenols	18	0 (−0.09, 0.1)	0.93	0.01	48.80
DPPH	14	0.04 (0.03, 0.06)	0.00	0.00	95.01
Anthocyanins	12	−0.25 (−0.58, 0.07)	0.13	0.67	0.00
Flavonoid	14	−0.06 (−0.12, 0.01)	0.10	0.76	0.00
Fruit Fresh Weight	13	0.32 (0.17, 0.47)	0.00	0.59	0.00
Fruit Dry Weight	8	0.22 (0.13, 0.31)	0.00	0.38	5.99
First Flowering Time	4	−0.01 (−0.06, 0.04)	0.60	0.30	17.24

## Data Availability

All data supporting the findings of this study are available in the paper.

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
