# Peer review of "Meta-Analysis of the Impact of Far-Red Light on Vegetable Crop Growth and Quality"

_plants, 2024, doi:10.3390/plants13172508_

Round 1

Reviewer 1 Report

Comments and Suggestions for Authors

The manuscript titled: Meta-analysis of the Impact of Far-red Light on Vegetable Crop Growth and Quality” by  Zhang and coworkers, submitted to Plants, demonstrates the influence of the far-red light on plant growth, in particular on horticultural crops.  

              The role of far-red light as a component of light spectrum, important for photosynthesis  and for photomorphogenetic processes, has been a subject of different papers for a long time and is quite well known. Since there are discrepancies or even controversial results of far-red light effects on plant growth, the authors performed comprehensive overview based on 207 independent studies from 55 literature sources published from 2014 to 2024 and analyzed the results.

Meta-analysis is a known tool to combine results from different studies, to enlarge samples for analysis and therefore obtain more reliable effect of particular agent, far-red light in this particular study. Therefore I consider the meta-analysis of already published results on the effect of far-red light on various growth parameters as an interesting task to reconcile different, even controversial data.

Methods are well performed and described. I appreciate careful analysis with the help of statistical methods.

The authors presented  graphically the effects of supplementary far-red light of different indicators from the number of scientific studies. The forest plots summarize information from individual studies showing on one hand heterogeneity of the results and on another - the estimated common effect.

The authors analyzed a) morphological indicators such as fresh weight, dry weight, and leaf area which showed positive effect of far-red treatment; b) physiological and biochemical indicators such as  malondialdehyde, relative chlorophyll content with negative effect, superoxide dismutase with positive effect, and activities of peroxidase and catalase without significant effect; c) nutritional indicators such chlorophyll content with negative impact, and the content of soluble sugar positively affected.

As an interesting side of this study I consider the idea of subgroup classification based on various factors. The subgroup analysis revealed significant differences in the effects of far-red  on different physiological indicators, such as far-red light intensity, plant species, or duration of light exposure.

I positively evaluate the idea of analysis, on a large scale, the effect of particular environmental factor, like far-red light in this study. However, the problem of discrepancies of the results due to different experimental and growth conditions and of course due to various plant species taken for the experiments, always remains. Because of response heterogeneity of different plant species in various experimental conditions each generalized effect is always  only an approximation.

Summarizing,  the authors established that far-red treatment significantly influence the morphological and nutritional indices of vegetables, it caused the increase of fresh weight, dry weight, plant height and soluble sugars content, while it caused the decrease of chlorophyll content and soluble protein. Furthermore, the authors pointed  that the effects of different physiological indicators such as far-red intensity, time of exposure and, the ratio of red light to far-red as well as plant species are important factors which should  be also taken into account during overall effect of far-red on vegetable crop growth and quality.

I highly evaluate “Meta-analysis of the Impact of Far-red Light on Vegetable Crop Growth and Quality” by  Zhang and coworkers and I recommend the manuscript to be published in Plants.

Author Response

#Reviewer:

Thank you for your advice on our work (Manuscript ID: plants-3178819). Each of your suggestions is very scientific and reasonable. This manuscript has been revised carefully according to your comments and suggestions. Because your bits of advice motivate us to constantly review and improve our manuscript. Our research results can be presented more clearly and reasonably because of your serious and responsible attitude. Thank you again for your advice, in our future work, we will definitely learn your scientific rigor attitude so that our scientific research work is more academically standardized. The replies to your comments are listed as follows:

Q1. However, the problem of discrepancies of the results due to different experimental and growth conditions and of course due to various plant species taken for the experiments, always remains. Because of response heterogeneity of different plant species in various experimental conditions each generalized effect is always only an approximation.

Answers: We particularly thank you for pointing out the issue of outcome heterogeneity in our study, which is a common issue when performing meta-analyses, and we consider this an important point of discussion.

In the meta-analysis, we have integrated data from multiple studies to assess the impact of a specific factor (in this study, far-red light). However, we acknowledge that differences in experimental conditions, plant species, and growth environments across studies may lead to heterogeneity in the results. To address this issue, we employed methods such as Subgroup Analysis, Sensitivity Analysis, Heterogeneity Assessment, Publication Bias Assessment, and the Random-Effects Model in our analysis. Yet, each generalized effect can still only be considered as an approximation. We believe that this approximation is an inherent characteristic of the meta-analytic method, allowing us to identify trends and patterns across a broad range of studies, even though these trends and patterns may be constrained by specific conditions.

We have continued to highlight these limitations in the discussion section and suggest that future studies should further explore the sources of this heterogeneity and how to reduce them through improved experimental design and standardized methods.

Reviewer 2 Report

Comments and Suggestions for Authors

The potential of far-red light (FR) to stimulate plant growth and enhance crop yield strongly interest researcher and growers, but their effects are complex as they strongly depend on other physiological and environmental factors. In this context, this meta-analysis on the effects of far-red light on growth and properties of vegetable crops is highly interesting and much needed. It is , well-done, well-conducted, and the potential biases well identified.

                I particularly appreciated the presentation of the FR effects on a large set of growth/physiological parameters (27, Table 1), and how the FR effects on five main responses (FW, DW, leaf area, stem diameter and Chl content) are modulated by four confounding factors (FR intensity, crop family, photoperiod-EOD, and R/FR) (figures 2-6). There are still important questions unanswered, notably how the overall PFD interacts with the R/FR, or the ratio between R/FR/Blue light. I assumed that there are too few studies that addressed these questions to be included in this meta-analysis. These questions should be mentioned in the Discussion to orient future research.

                Before its publication, I recommend the following improvements/clarifications.

1- In the Introduction, the authors mentioned that “FR is primarily involved in the photoperiodic effect” (li. 36-37) via the phytochrome. This is true but maybe the FR effects are in a larger part the consequence of the shade avoidance responses mediated by the phytochrome under low R/FR ratio. Generally, shade avoidance response under high FR levels results in higher plants (longer internodes), larger leaves (and longer petioles), which increase light interception efficiency.

2- The structure of the manuscript must be improved. There are two subsections (2.2 and 2.3)of the Results section with the same subtitle (Subgroup analysis), this is confusing. Also, the results in Tables 2 and 3 (incorrectly indicated as 3.2 and 3.3 in line 301) are presented only in the Discussion section. This again is unexpected and confusing for the reader.

3- Also, the structure will be clearer by changing in the sentence of line 132-133 “…to examine the impacts of different factors (FR intensity, crop family, photoperiod-EOD, and R/FR) on the parameters fresh weight, …”.

4- In line 223, I guess we should read Figure 2 and 4, instead of 5 and 7.

5- The quality of the text is good, but he sentence in line 42-43 must be improved. “When plants are exposed to red light, the Pr form is converted to the Pfr form, and FR promotes the Pfr reversion to Pr.”

Comments on the Quality of English Language

See minor details in the comments above.

Author Response

#Reviewer:

Thank you for your advice on our work (Manuscript ID: plants-3178819). Each of your suggestions is very scientific and reasonable. This manuscript has been revised carefully according to your comments and suggestions. Because your bits of advice motivate us to constantly review and improve our manuscript. Our research results can be presented more clearly and reasonably because of your serious and responsible attitude. Thank you again for your advice, in our future work, we will definitely learn your scientific rigor attitude so that our scientific research work is more academically standardized. Based on the instructions provided in your letter, revisions in the text are shown using highlight for additions. The replies to your comments are listed as follows:

Q1. In the Introduction, the authors mentioned that “FR is primarily involved in the photoperiodic effect” (li. 36-37) via the phytochrome. This is true but maybe the FR effects are in a larger part the consequence of the shade avoidance responses mediated by the phytochrome under low R/FR ratio. Generally, shade avoidance response under high FR levels results in higher plants (longer internodes), larger leaves (and longer petioles), which increase light interception efficiency.

Answers: Thank you for pointing out this. We agree that under low R/FR ratio conditions, the activation of phytochromes leads to a series of morphological changes in plants, which helps them gain an advantage in light competition. We have revised the introduction section. (line 41 - 51)

Q2. The structure of the manuscript must be improved. There are two subsections (2.2 and 2.3)of the Results section with the same subtitle (Subgroup analysis), this is confusing. Also, the results in Tables 2 and 3 (incorrectly indicated as 3.2 and 3.3 in line 301) are presented only in the Discussion section. This again is unexpected and confusing for the reader.

Answers: Thank you for pointing out this. We have modified the manuscript according to the comment. Tables 2 and 3 present the subgroup analysis results for the soluble sugar and soluble protein content, respectively. It is true that there was insufficient description in the results section, which has been supplemented in sections 2.2.1 to 2.2.4 (line 135 - 193, line 224 and 237).

Q3. Also, the structure will be clearer by changing in the sentence of line 132-133 “…to examine the impacts of different factors (FR intensity, crop family, photoperiod-EOD, and R/FR) on the parameters fresh weight, …”.

Answers: Thank you for pointing out this. We have modified the manuscript according to the comment (line 136 and 137).

Q4. In line 223, I guess we should read Figure 2 and 4, instead of 5 and 7.

Answers: Thank you for pointing out this. We have modified the manuscript according to the comment.

Q5. The quality of the text is good, but he sentence in line 42-43 must be improved. “When plants are exposed to red light, the Pr form is converted to the Pfr form, and FR promotes the Pfr reversion to Pr.”

Answers: Thank you for pointing out this. We have modified the manuscript according to the comment (line 43 and 44).
